# Rural Industrial Integration’s Impact on Agriculture GTFP Growth: Influence Mechanism and Empirical Test Using China as an Example

**DOI:** 10.3390/ijerph20053860

**Published:** 2023-02-21

**Authors:** Yafei Wang, Huanhuan Huang, Jing Liu, Jin Ren, Tingting Gao, Xinrui Chen

**Affiliations:** School of Economics and Management, Chongqing Normal University, Chongqing 401331, China

**Keywords:** rural industrial integration, agriculture GTFP, rural human capital, rural land transfer

## Abstract

Agricultural carbon emission is an significant cause of global climate change and many environmental and health problems. Achieving low-carbon and green development in agriculture is not only an inevitable choice for countries around the world to cope with climate change and the accompanying environmental and health problems, but also a necessary path for the sustainable development of global agriculture. The promotion of rural industrial integration is a practical way to realize sustainable agricultural growth and urban–rural integration development. The analysis framework of agriculture GTFP is creatively extended in this study to include the integration and growth of rural industries, rural human capital investment and rural land transfer. According to the sample data of 30 provinces in China from 2011 to 2020 and the systematic GMM estimation method, and through the combination of theoretical analysis and empirical testing, this paper discusses the influence mechanism of rural industrial integration development on agriculture GTFP growth, as well as the regulating role of rural human capital investment and rural land transfer. The results show that rural industrial integration has significantly promoted the growth of agriculture GTFP. Additionally, after decomposing agriculture GTFP into the agricultural green technology progress index and agricultural green technology efficiency index, it is found that rural industrial integration has a more obvious role in promoting agricultural green technology progress. Furthermore, quantile regression found that with the increase in agricultural GTFP, the promoting effect of rural industrial integration showed an “inverted U-shaped” feature. Through heterogeneity testing, it is found that the agriculture GTFP growth effect of rural industrial integration is more obvious in areas with high level of rural industrial integration. Additionally, as the nation places more and more focus on rural industrial integration, the promotion role of rural industrial integration has become more and more obvious. The moderating effect test showed that health, education and training, migration of rural human capital investment and rural land transfer all strengthened the promoting effect of rural industrial integration on agricultural GTFP growth to varying degrees. This study provides rich policy insights for China and other developing countries around the world to address global climate change and many related environmental and monitoring issues by developing rural industrial integration, strengthening rural human capital investment and promoting agricultural land transfer to achieve sustainable agricultural growth and reduce undesirable output outputs such as agricultural carbon emissions.

## 1. Introduction and Literature Review

Environmental issues brought on by global climate change have emerged as a serious threat to human society and a significant practical concern that needs to be addressed by governments all over the world. Item 13 of the United Nations 2030 Agenda for Sustainable Development (SDGs) set up in 2015 was “Climate Action: Urgent Action to address climate change and its Impacts” (SDG 13), the main task of which is to mitigate the impacts of climate change on mankind and improve the capacity to cope with climate change. The key is to tackle economic, social and environmental problems in a coordinated way. The trend of high agricultural carbonization has become increasingly evident in recent years, and has become an important contributor to global climate change and a significant barrier to the realization of sustainable development in the developing countries represented by China. Under the trend of multiple goals such as economic catch-up and poverty alleviation, developing countries have widespread practical problems such as overexploitation and extensive utilization of agricultural resources. [1]. According to the statistics of the Food and Agriculture Organization of the United Nations (FAO) in 2020, agricultural land releases more than 30% of the global total man-made greenhouse gas emissions, equivalent to the production of 15 billion tons of carbon dioxide every year. The low-carbon green development in agriculture is not only an inevitable choice for the world to cope with climate change and other accompanying environmental and health problems, but also a necessary path for sustainable development of global agriculture. 

GTFP is an effective way to promote low-carbon and green agricultural development [2,3]. GTFP originates from the basic idea of GTFP proposed by Solow [4], which has been widely applied in the academic circle to measure the quality of economic growth or development [5,6]. TFP describes the growth degree of “desirable output” driven by innovation or management, such as technological progress and allocation efficiency improvement, excluding tangible factors such as labor and capital. However, it does not include “undesirable output” caused by environmental pollution in the measurement framework of economic growth performance. GTFP not only reflects the efficiency of factor input into output in the process of economic development under the condition that labor, capital and other input factors are given, but also incorporates desirable output and undesirable output such as carbon emission into the accounting framework. It can represent the coordinated and sustainable level of economy and environment of a country or region [7,8,9].

As the world’s largest developing country, China has adopted an extensive growth model of “high input and high output” to address food security and rural poverty. While promoting agricultural productivity and increasing farmers’ income, it has also led to the deterioration of the rural ecological environment, rapid growth in agricultural carbon emissions, low agricultural production efficiency and other problems, posing serious challenges to sustainable agricultural development. Promoting the growth of agricultural GTFP in order to achieve the quality change, efficiency change and power change of agricultural development has become an important orientation for the Chinese government to implement agricultural policies at present and even for a long period of time in the future. The research on the driving factors and implementation mechanism of agricultural GTFP has become a long-term, continuous and highly theoretical and practical problem.

The existing literature has explored the impact mechanism of agricultural GTFP growth from different perspectives, such as resource allocation distortion [10], fiscal support for agriculture [11], agricultural foreign direct investment [12], infrastructure construction [13] and urbanization process [14,15]. However, few works in the literature discuss the impact mechanism of rural industrial integration on agricultural GTFP growth from the perspective of the change in industrial form. In fact, under the joint action of the extensive application of modern science and technology to agriculture or rural areas and the profit-seeking drive of relevant agricultural operating subjects, the internal agricultural industry (agriculture, forestry, livestock, subfishery) and between agriculture and the secondary and tertiary industries also shows an increasingly obvious trend of integration, which is mainly manifested as follows. With agriculture as the industrial base, modern agricultural operation subject with moderate scale operation as the core, interest linkage mechanism among related subjects as the link, and vertical extension of agricultural industry chain, multifunctional expansion of agriculture, integration of agricultural service industry and cultivation of new agricultural forms as the means, a new industrial development model of factor resource integration, value chain interpenetration and industrial cross-coordination between agriculture and rural secondary and agricultural industries emerges [16,17,18]. In addition, in practice, the integrated development of rural industries has been highly valued by the governments of various countries. For example, the Japanese government adopted the “six industries” thought proposed by Professor Naratomi Imamura in 1996, introduced the “Law on the Promotion of Agriculture and Industry” in 2008 and further put forward the guidelines for vigorously developing “six industries” in 2013. It aims to promote the rapid integration of agricultural production into processing, circulation and sales, so that the domestic agriculture can take the initiative in international competition. The South Korean government actively promotes the integrated development of rural industries, fully integrates human, natural and cultural resources in rural areas, promotes the development of regional agriculture and stimulates the vitality of rural areas by increasing the added value of agricultural industry, so as to break through the limitations of simple agricultural production and essentially change the mode of agricultural development. The French government vigorously supports farmer cooperatives and other agricultural operation organizations, and promotes them to extend the agricultural industry chain, achieve deep integration with the agricultural processing industry and service industry and maximize the added value and factor utilization efficiency of the agricultural industry.

Taking China as an example, this study mainly discusses the coordination between economy and environment in the agricultural field from the perspective of rural industry integration, incorporates agricultural carbon emission, an undesirable output, into the agricultural total factor productivity measurement framework, calculates the agricultural GTFP growth index an discusses the impact of rural industry integration on agricultural GTFP. Specifically, this study firstly discusses the influence mechanism of rural industry integration on the growth of agricultural GTFP from the theoretical level. Secondly, taking China as an example, the agricultural GTFP index and rural industry integration index of 30 sample provinces in China from 2011 to 2020 were measured, respectively. Thirdly, the systematic GMM estimation method was used to empirically test the influence and effect of rural industry integration on agricultural GTFP, as well as the heterogeneity in different regions and different time periods. In addition, this study also discusses the moderating role of rural human capital investment, such as health, education and training, migration and rural land transfer.

The primary contribution of this study, in comparison to earlier ones, is the inclusion of rural industry integration for the first time in the context of agricultural GTFP analysis, as well as the theoretical justification and empirical evaluation of the GTFP growth effect of rural industry integration. Additionally, it presents numerous scenarios and mechanisms of rural industry convergence affecting agricultural GTFP based on the heterogeneity test of different times and spaces, as well as the moderating roles of rural human capital investment and rural land transfer in the growth effect of rural industry convergence on GTFP. Through fostering rural industrial integration, boosting rural human capital investment, and increasing agricultural land circulation, especially for developing countries worldwide, this study offers China valuable experience and rich policy implications for achieving sustainable agricultural growth.

## 2. Theoretical Analysis and Research Hypothesis

### 2.1. Influence Mechanism of Rural Industrial Integration on Agricultural GTFP

Agricultural TFP growth is mainly expressed as the improvement of input–output efficiency triggered by technology progress and resource allocation efficiency in agricultural production and operation; thus, agricultural GTFP growth also covers the reduction in environmental pollution, a undesirable output level. The impact of rural industrial integration on the growth of agricultural GTFP is mainly reflected in three aspects, including the effect of technology progress, factor reallocation and ecological environment optimization.

(1)Technology progress effect. Through geographic proximity, talent flow and technical interaction between agricultural and related business entities, rural industrial integration realizes the deconstruction, reorganization and extension of industrial chains between secondary and tertiary industries and agriculture. This improves the technical level of agricultural production and encourages the overflow of advanced technology and management experience from non-agricultural industries to agriculture [19]. The new industries and models derived from rural industrial integration, such as ecological agriculture, recycling agriculture and intelligent agriculture, also have high technical ability and advanced process management mode. Taking intelligent agriculture as an example, the development of agricultural “intelligence” has improved the technical capacity of all aspects of agriculture and realized the development of agricultural precision, intelligence and intensification.(2)Factor reallocation effect. Rural industrial integration has improved the conditions of agricultural factor endowment and increased the efficiency of agricultural factor allocation through the effective integration of urban secondary and tertiary industries. It has also strengthened the connection between urban and rural industries and prompted the diffusion and penetration of production factors such as technology, capital, talents, management and information into the field of agricultural industries [20,21]. Through aggregation, penetration and cross-reorganization among the primary, secondary and tertiary industries, rural industrial integration redistributes rural capital, technology and resources across borders and realizes the flow and full interaction of capital, technology, talent, information, management and other elements in this process. Additionally, factor system integration encourages the optimum allocation of diverse production factors in deeper fields and at higher levels, which significantly enhances the effectiveness of agricultural factor allocation. Rural industrial integration not only creates new forms of business, but also breaks through the traditional function of supplying agricultural products, promotes the multi-functional development of production, service, ecology and society of agriculture, maximizes the potential of converting agricultural resources into economic value and promotes the comprehensive application of resource elements and the maximization of output value [22,23].(3)Ecological environment optimization effect. Agricultural information and technology services extended by the rural industrial integration, such as information or intelligent management and remote sensing technology, improve the level of agricultural production technology, optimize the agricultural production business process, reduce the traditional human, material and chemical fertilizer and pesticide resource consumption in the agricultural production and operation process and help reduce agricultural ecological pollution [24,25]. Consider eco-agriculture, one of the forms of industrial integration, which combines traditional agriculture with cutting-edge ecologically sound technology. Eco-agriculture emphasizes not only the full utilization of agricultural resources but also the scientific conservation and restoration of agricultural resources and ecosystems, producing safe and healthy agricultural products while also fostering the improvement of the rural ecological environment. Another example is circular agriculture, which establishes a system of reciprocal conditions, mutual utilization, and perpetuation of production factors among various agricultural segments, realizing the reduction in waste emission in the production process, or even zero emission and resource reuse, and thereby reducing the use of pesticides, veterinary drugs, chemical fertilizers and conventional energy, which forms a production pattern of clean production, low input, low consumption, low emission and high efficiency, and improves the comprehensive allocation efficiency of agricultural resources and ecological environment quality.

### 2.2. The Regulating Role of Rural Land Transfer and Rural Human Capital Investment

#### 2.2.1. The Regulating Role of Rural Land Transfer

The land fragmentation management mode and land resource allocation mode with low-income farming family households as the basic unit are difficult to combine with the moderately large-scale, intensive and specialized land management attributes contained in the development of modern agriculture, which is an important cause of the misallocation and efficiency loss of agricultural production factors such as land, labor, capital and technology [26]. Transferring the fragmented elements currently dispersed among families to farmers’ cooperatives, family farms, large farming and breeding households and other modern agricultural management organizations can significantly increase the level of agricultural mechanization and the application of modern agricultural technology equipment and management methods, and promote the advancement of agricultural technology and the comprehensive allocation efficiency of factors [27]. The development of modern agricultural management organizations resulting from land transfer helps to better absorb modern agricultural technology, production equipment and management methods, and also helps agricultural management entities to embed into various links of modern agriculture and related industrial chains, participate in specialized division of labor and market-oriented collaboration, and enhance the efficiency of agricultural production and operation [28]. In addition, the moderate scale agricultural operation organization under land transfer has a stronger ability to predict and dispose agricultural market and natural risks, as well as the ability to acquire and allocate financial resources, which also contributes to the expansion of agricultural reproduction and the improvement of scale efficiency. In the process of rural industrial integration acting on the growth of agricultural GTFP, if the scale of land transfer is low, it will restrict the cultivation and development of modern agricultural management organizations. Considering that compared with modern agricultural management organizations, traditional low-income farming family households have multiple difficulties, such as low market acuity, low decision-making efficiency, lack of factor allocation ability and so on, it is difficult to fully tap into or enjoy the multiple growth dividends released by rural industry integration, which weakens the promoting effect of rural industry integration on the growth of agricultural green total factor productivity. On the contrary, it has a strengthening effect. Therefore, we propose hypothesis 2 to be tested: rural land transfer plays a positive regulating role in the impact of rural industrial integration on agricultural GTFP.

#### 2.2.2. The Regulating Role of Rural Human Capital Investment

Among many production factors, human capital is the most creative and active production factor. It is also the main factor of allocating land, material capital, technology, information and other production factors. Since Schultz [29] put forward the concept, human capital has become an important perspective for new growth theories to interpret the source and difference of productivity growth. Relevant studies have confirmed that healthy human capital investment can improve workers’ health level, promote workers’ long-term production and operational investment, and improve the long-term operating performance of economic activities [30]. Investment in education and training is an important way for workers to acquire knowledge and vocational skills, which helps to improve the efficiency of decision-making and business performance of economic activities and to increase labor productivity. Additionally, by investing in transportation and communication, this can increase social capital, expand the economic and personal space of workers, and support innovative, entrepreneurial and high-quality employment activities [31,32]. Investment in rural health, education and training, and transportation and communication human capital can improve the overall human capital level of rural workers or production and business entities, which is an important contributor to agricultural technology progress and technology efficiency improvement. Among the influences of rural industrial integration on agricultural GTFP growth, higher rural human capital level can effectively capture the spillover effect of rural industrial integration on agricultural technology progress and technology efficiency improvement, and then promote the growth of agricultural GTFP. Based on the above analysis, we propose hypothesis 3 to be tested: rural human capital plays a positive regulating role in the impact of rural industrial integration on agricultural GTFP, Figure 1.

## 3. Models, Estimation Methods and Variables

In order to explore the impact of rural industry integration on the growth of agricultural GTFP, we first established a benchmark model, and included one stage lag of agricultural GTFP in the explanatory variable of the model, which can not only describe the “inertia” or path dependence of agricultural GTFP itself, but also alleviate the deviation of the estimated results caused by missing variables. The specific model is set as Equation (1). For dynamic panel models, the improved system GMM method based on difference GMM can greatly alleviate the endogenous problem of the model and improve the robustness of parameter estimation [33,34]. Based on the difference of the selection of weight models, the GMM estimation of the system is divided into one-step and two-step estimation. Under normal circumstances, the standard covariance matrix of the two-step estimation method can deal with sequence autocorrelation and heteroscedasticity more effectively, and the estimation effect is more robust [35,36]. Therefore, we use the system GMM estimation method to estimate the parameters of Equation (1).
(1)$$GTFP_{it}=\alpha +\beta_{1}GTFP_{it-1}+\beta_{2}RII_{it}+\beta_{3}X_{it}+\mu_{i}+\lambda_{t}+\varepsilon_{it}$$

In Equation (1), RII is the core explanatory variable, representing the level of integration of the three rural industries. GTFP is the explained variable, representing agriculture GTFP. X represents the vector of control variables, and i and *t* represent the i province and the t period, respectively. $\mu_{i}$ denotes area fixed effects, $\lambda_{t}$ denotes year fixed effects and $\varepsilon_{it}$ denotes the random perturbation term.

In addition, in order to further discuss the moderating effect of rural human capital and rural land transfer on the influence of rural industrial integration on the growth of agricultural GTFP, we add the interaction terms between rural industrial integration and rural human capital, and the interaction terms between rural industrial integration and rural land transfer, respectively, into the benchmark Equation (1) to obtain Equation (2) and Equation (3).
(2)$$GTFP_{it}=\alpha +\beta_{1}GTFP_{it}+\beta_{2}RII_{it}+\beta_{3}X_{it}+\beta_{4}HC_{it}+\beta_{5}(RII_{it}\times HC_{it})+\mu_{i}+\lambda_{t}+\varepsilon_{it}$$
(3)$$GTFP_{it}=\alpha +\beta_{1}GTFP_{it}+\beta_{2}RII_{it}+\beta_{3}X_{it}+\beta_{4}CIR_{it}+\beta_{5}(RII_{it}\times CIR_{it})+\mu_{i}+\lambda_{t}+\varepsilon_{it}$$

In Equation (2) and Equation (3), $HC_{it}$ and $CIR_{it}$ represent rural land transfer and rural human capital investment, respectively. $HC_{it}$ will be subdivided into three categories, MH (migratory human capital), EH (educational human capital) and HH (healthy human capital), in the later empirical tests. In addition to simplifying the model and avoiding too many parameters that lead to an unrecognizable model, the test process contains only one interaction term at a time, and the test is performed sequentially. For the parameter estimation of both Equation (2) and Equation (3), we used a two-step systematic GMM approach.

### 3.1. Variables

#### 3.1.1. Explained Variable

The measurement of production efficiency by data enveloping analysis (DEA) is prone to result deviation due to different radial and angle choices. In order to eliminate this deviation, Tone Karou (2001) [37] introduced the relaxation variable into the objective function and proposed a slack-based measure (*SBM*) non-radial and non-angular efficiency measurement methods, which have been widely used in productivity measurement field in recent years. The Malmquist–Luenberger (*ML*) index method and the global Malmquist–Luenberger (GML) index method are widely used in the calculation and decomposition of GTFP [38,39,40,41,42]. Considering the practicability of operation and the objective authenticity of measurement, this study adopted the SBM-ML index method to measure the agricultural GTFP in 30 provincial regions of China. In this paper, each province is taken as a decision-making unit, and each decision-making unit has three elements: There are *N* inputs $X=\left\{x_{1},x_{2},\ldots ,x_{N}\right\}\in R_{+}^{N}$, *Q* desirable outputs $Y=\left\{y_{1},y_{2},\ldots y_{Q}\right\}\in R_{+}^{Q}$ and *L* undesirable output $B=\left\{b_{1},b_{2},\ldots ,b_{L}\right\}\in R_{+}^{L}$ in agricultural production in each province. Assuming variable returns to scale, then the directional distance function of SBM is:(4)$$D_{V}^{t}\left(x_{i}^{t},y_{i}^{t},b_{i}^{t}\right)=\overset{⌢}{p}=\min\frac{1-\left[\frac{1}{N}\sum_{n=1}^{N}\frac{s_{n}^{x}}{x_{n}^{i}}\right]}{1+\left[\frac{1}{Q+L}(\sum_{q=1}^{Q}\frac{s_{q}^{y}}{y_{q}^{i}}+\sum_{l=1}^{L}\frac{s_{l}^{b}}{b_{l}^{i}})\right]}$$
(5)$$s.t.\left\{\begin{array}{l} \sum_{i=1}^{I}z_{i}^{t}y_{i,q}^{t}-s_{q}^{y}=y_{i,q}^{t},q=1,2,\ldots ,Q;\\ \sum_{i=1}^{I}z_{i}^{t}x_{i,n}^{t}+s_{n}^{x}=x_{i,n}^{t},n=1,2,\ldots ,N;\\ \sum_{i=1}^{I}z_{i}^{t}b_{i,l}^{t}+s_{l}^{b}=b_{i,l}^{t},l=1,2,\ldots ,L;\\ \sum_{i=1}^{I}z_{i}^{t}=1,z_{i}^{t}\geq 0,s_{q}^{y}\geq 0,s_{n}^{x}\geq 0,s_{l}^{b}\geq 0,i=1,2,\ldots I \end{array}\right.$$

In Equation (5), $\overset{⌢}{p}$ is the efficiency evaluation index, $x_{i}^{t}$ represents the input of i province, $y_{i}^{t}$ represents the desired output of i province, $b_{i}^{t}$ represents the undesired output of i province, $s_{n}^{x}$ represents the overinput, $s_{q}^{y}$ represents the insufficiency of the desired output, $s_{t}^{b}$ represents the redundancy of the undesired output, and $z_{i}^{t}$ represents the weight vector.
(6)$$\begin{array}{l} {\left(SBM-ML\right)}_{t}^{t+1}={\left[\frac{D_{V}^{t}\left(x^{t+1},y^{t+1},b^{t+1}\right)}{D_{V}^{t}\left(x^{t},y^{t},b^{t}\right)}\times \frac{D_{V}^{t+1}\left(x^{t+1},y^{t+1},b^{t+1}\right)}{D_{V}^{t+1}\left(x^{t},y^{t},b^{t}\right)}\right]}^{1/2}\\ =\frac{D_{V}^{t+1}\left(x^{t+1},y^{t+1},b^{t+1}\right)}{D_{V}^{t}\left(x^{t},y^{t},b^{t}\right)}\times {\left[\frac{D_{V}^{t}\left(x^{t+1},y^{t+1},b^{t+1}\right)}{D_{V}^{t+1}\left(x^{t+1},y^{t+1},b^{t+1}\right)}\times \frac{D_{V}^{t}\left(x^{t},y^{t},b^{t}\right)}{D_{V}^{t+1}\left(x^{t},y^{t},b^{t}\right)}\right]}^{1/2}\\ =Ec_{t}^{t+1}\times Tc_{t}^{t+1} \end{array}$$

Equation (6) is the adjacent reference SBM-ML index of T to period T + 1. The SBM-ML index can be decomposed into EC (technical efficiency change index) and TC (technical progress change index), so when SBM-ML > 1, EC > 1, TC > 1, it means the improvement of agricultural GTFP, the increase in technical efficiency, and the progress of technology, respectively. SBM-ML < 1, EC < 1 and TC < 1, respectively, indicate the decrease in agricultural GTFP, technical efficiency and technical regression.

The specific input and output indicators of this study are described as follows:

(1) Agricultural inputs, including capital and labor inputs and other productive materials or resources consumed in addition. ① Capital investment. The total agricultural fixed capital formation and corresponding price indices at the provincial level in previous years cannot be found in the officially published statistics, so capital accounting is not possible. Therefore, most of the existing literature does not include agricultural capital input when measuring the TFP in agriculture. Considering that agricultural capital is still an indispensable factor in promoting agricultural economic growth, and that in recent years, the state has been increasing its investment in fixed assets in agriculture through fiscal leverage to reverse the predicament of net capital outflow from agriculture or rural areas, which obviously has an important role in promoting the growth of the TFP in agriculture, ignoring this factor will undoubtedly lead to an underestimation of the TFP in agriculture and will also exaggerate the contribution of other input factors [43]. In this study, the total amount of fixed investment in agriculture, forestry, animal husbandry and fishery by provinces in previous years is approximated instead of capital input and treated in constant 2000 prices to exclude the effect of price factors, unit (CNY billion). ② Labor input. In order to effectively reflect the actual input of agricultural labor in a certain period, this study uses the number of employees in the primary industry at the end of the year to represent the agricultural labor input. Due to the missing data of the years 2011 in Anhui and 2011–2013 in Heilongjiang, the mean interpolation method was used to make up the values considering the exponential growth of the population. When interpolating with stata14.0, the logarithm was taken first, then the mean method was interpolated, and finally the logarithm was taken again, which is closer to the real value of the missing data, units (million). ③ Production means or resource consumption. First, the area sown for crops, an indicator that better reflects the actual degree of land use compared to the area of cultivated land, unit (thousand hectares). Usually, the effective irrigation area should be equal to the sum of watered land area in paddy fields and drylands where irrigation equipment has been supported and normal irrigation can be carried out, which is an important indicator reflecting the degree of hydrologization in each region of China, unit (thousand hectares). Third, the total power of agricultural machinery refers to the total power of machinery used in agriculture, forestry, animal husbandry and fishery production. It is the specific composition of the total power of fishery machinery + total power of combine harvesters + power of agricultural drainage and irrigation diesel engines + the sum of power of agricultural drainage and irrigation motors, compared to the total number of machinery. This indicator better reflects the actual resources consumed by agricultural production, unit (million kilowatts). Fourth, fertilizer inputs, expressed in terms of nitrogen fertilizer, phosphate fertilizer, potash fertilizer and compound fertilizer application discount consumed in agricultural production, unit (million tons).

(2) Agricultural output, including desired and undesirable output output. ① Desired output. Expressed as total output value of agriculture, forestry, animal husbandry and fishery. To exclude the effect of prices, this study used the price index of the total output value of agriculture, forestry, animal husbandry and fishery (2011 = 100) for the total output value of agriculture, forestry, animal husbandry and fishery. ② undesirable output output. The undesirable output output of agriculture is mainly reflected in the agricultural carbon emissions caused by six major factors such as fertilizer, pesticide, agricultural film, diesel, tillage and irrigation [44,45]. Therefore, this study uses agricultural carbon emissions as a proxy variable for undesirable output output, unit (million tons). The accounting formula is as follows:
(7)$$E=\sum_{}^{n}E=T_{i}\times \sigma_{i}$$

In Equation (7), *E* is the total carbon emissions from agricultural production activities; *E_i_* represents emissions from all types of carbon sources. *i*, *n* characterize the i carbon source to the n carbon source. *T_i_* represents the original amount of carbon emissions from each source. σ_i_ represents the emission factor for each carbon emission source. The basis of determination is shown in Table 1 below.

The data of agricultural input and output indicators in this study are obtained from the China Rural Statistical Yearbook, the official website of the National Bureau of Statistics. Considering the high data sensitivity requirement of MAXDEA software and the different units and great data differences among the selected variables, the input and output data are dimensionless in this paper to eliminate the differences in data of different magnitudes.

#### 3.1.2. Explanatory Variable

The explanatory variable in this study is the rural industrial integration development index. At present, this development index is not officially published, and this study starts from the connotation of rural industrial integration and refers to the research results of Wang (2020) [50], and Zhang and Li (2022) [51]. Based on the comprehensiveness of indicators and the availability of data, eight secondary indicators in four dimensions, such as extension of agricultural industry chain, expansion of agricultural multi-functionality, integration of agricultural service industry and penetration of agricultural technology, are selected according to the principles of scientific, systematical and hierarchical nature to construct a comprehensive index system for evaluating the development of rural industrial integration, as shown in Table 2.

First, agricultural industry chain extension. The extension of agricultural industry chain refers to the vertical extension of agricultural industry to pre-production and post-production through the linkage and integration of agricultural product production, processing and marketing. Vigorous development of the agricultural product processing industry to improve the added value of agriculture is the main direction of agricultural industry chain extension, and farmers’ professional cooperatives are the main organizational carrier for establishing vertical division of labor and cooperation in the agricultural industry chain. Therefore, this study selects two indicators, the proportion of agricultural processing industry and the scale of farmers’ professional cooperatives, to measure the level of agricultural industry chain extension.

Second, multifunctional expansion of agriculture. Multifunctional expansion of agriculture refers to the continuous expansion of agricultural economic, social, cultural and ecological functions, promoting the deep integration of agriculture with rural tourism, culture, recreation and other industries, forming new business model and new modes of integrated development of rural industries, which brings diversification of rural employment channels and income sources. Therefore, three indicators, including the proportion of leisure agriculture, the level of facility agriculture and the proportion of rural non-farm employment, are selected to portray the degree of multifunctional expansion of agriculture.

Third, agricultural services integration. The integration of agricultural service industry refers to the integration, interaction and coordinated development of agriculture and agricultural service industry, which provides services before, during and after production of agriculture. There are various types of agricultural services, including agricultural machinery operation service, agricultural technology extension service, agricultural materials distribution service, agricultural information service, agricultural products sales service, etc. In this study, the proportion of agriculture, forestry, husbandry and fishery services was selected to measure the level of integration of agricultural services.

Fourth, agricultural technology penetration. Agricultural technology penetration refers to the extent to which agricultural equipment or technologies such as agricultural machinery and equipment and modern communications are introduced and applied in the field of agricultural production, with the aim of improving the level of technology and labor productivity in the process of agricultural production and operation. Therefor, agricultural mechanization and agricultural labor productivity were used to measure the level of agricultural technology penetration.

The entropy method was used to measure the rural industrial integration development index by drawing on Li (2022) [52]. Firstly, the data are unified using the extreme difference standardization method to eliminate the influence of the magnitude. Since the indicators are all positive, they are standardized as:(8)$${Y^{\prime }}_{ij}=\frac{Y_{ij}-\min Y_{ij}}{\max Y_{ij-}\min Y_{ij}}$$

In this equation, Y is the standardized value of i provinces’j indicators, and the weight of each indicator after standardization is calculated as $P_{ij}$:(9)$$P_{ij}={Y^{\prime }}_{ij}/\sum_{i=1}^{n}{Y^{\prime }}_{ij}$$

Then, the information entropy $E_{j}$, coefficient of variation $g_{j}$ and weight $W_{j}$ of the j indicator are calculated: (10)$$E_{j}=-\ln{\left(n\right)}^{-1}\sum_{i=1}^{n}P_{ij}\ln P_{ij},g_{j}=1-E_{j},\;W_{j}=\frac{g_{j}}{\sum_{j=1}^{n}g_{j}}$$

Finally, the rural industrial integration development index $F_{j}$ is calculated: (11)$$F_{i}=\sum_{j=1}^{n}W_{j}\times {Y^{\prime }}_{ij}$$

#### 3.1.3. Control Variables

To minimize the bias caused by omitted variables, the following control variables were selected for this study based on the current literature [53,54,55]. (1) The level of urbanization, expressed as the share of the resident urban population in the total population of each province; (2) the degree of openness to the outside world, expressed as total agricultural exports and imports divided by regional GDP; (3) the level of economic development, expressed as net income per rural resident, and the unit is CNY; (4) the financial support for agriculture, whose inputs mainly include support for agricultural production expenditures, agricultural machinery purchase subsidies, direct grain subsidies, comprehensive subsidies for agricultural materials and business expenses of the agriculture, forestry, water and meteorological departments. Considering that the level of agricultural development varies between provinces and municipalities, the intensity of financial support for agriculture is expressed by using the percentage of financial support for agriculture as a percentage of the total output value of agriculture, forestry, animal husbandry and fishery; (5) the degree of industrialization, expressed as the share of gross domestic product of the secondary sector in the gross regional product; (6) the digital inclusive finance, expressed by the digital inclusive finance development index of each province publicly released by the Digital Finance Research Center of Peking University.

#### 3.1.4. Regulating Variables

(1) Rural human capital investment, broken down into three specific categories: migration human capital investment, represented by rural transportation and communication expenditures in each province; education human capital investment, represented by rural education and recreation expenditures in each province; and healthy human capital investment, represented by rural health care expenditures in each province. (2) Land transfer, expressed as the proportion of the transferred area of farmland contracted by rural families to the total area of farmland contracted for operation in each province.

#### 3.1.5. Data Sources and Descriptive Statistics

This study collects panel data of 30 provinces and municipalities directly under the central government and autonomous regions in China from 2011 to 2020, except for Tibet, Hong Kong, Macao and Taiwan, from the China Statistical Yearbook, China Rural Statistical Yearbook and statistical yearbooks of provinces and municipalities directly under the central government and autonomous regions. The data of all price-related variables are deflated by 2011 as the base period, and the data of large indicators are taken as logarithms. Descriptive statistics for each variable are shown in Table 3.

## 4. Empirical Testing

### 4.1. Baseline Regression

The generalized moment estimation method of the system does not need to assume variable distribution and know the distribution of random disturbance terms to effectively solve the endogeneity problem [56]. In this study, the system generalized method of moments (system GMM) is used to estimate the parameters of Equation (1). The estimated results are shown in Table 4, where column (1) is the estimated results with variable region and time effects. Column (2) controls for the regional effect without controlling for the time effect, column (3) controls for the time effect without controlling for the regional effect, and column (4) controls for both the regional and the time effect. In this study, the regression results of column (4) were discussed. In column (4) of Table 4, AR(1) is less than 0.1, AR(2) is greater than 0.1 and the *p*-value of Hansen test is greater than 0.1, which satisfies the prerequisite for using the generalized moment estimation of the system, that is, the residual sequence in the difference model only has first-order autocorrelation, but not second-order and higher-order autocorrelation, and the instrumental variables have strict exogeneity. In conclusion, it can be preliminarily judged that the estimation results of the generalized moment estimation of the system are consistent and reliable.

As can be seen from column (4), the coefficient of the explanatory variable lagged by one period GTFP_it-1_ is significantly positive, indicating that the growth of agricultural GTFP has a more obvious path-dependent feature. The coefficient of the effect of rural industrial integration (RII) is significantly positive at the 5% level, indicating that the development of rural industrial integration helps to improve agricultural green agriculture GTFP, and each unit increase in rural industrial integration will drive agricultural green agriculture GTFP growth by 0.243 units.

For the other control variables: The level of urbanization (URB), financial support to agriculture (FINA), industrialization level (INDU), openness to the outside world (OPEN), and digital inclusive finance (DIF) all have significantly positive coefficients. This indicates that the control variables are significantly contributing to promoting the growth of green growth rate in agriculture.

### 4.2. Robustness Tests

#### 4.2.1. Robustness Test Based on Quantile Regression

The basic idea of quantile regression is mainly derived from Koenker and Bassett (1978) [57], and further studies were conducted by Koenker and Hallock (2001) [58], which mainly focused on the influence of various variables on samples at different subsites of conditional distribution. The sensitivity of quantile regression results to outliers can be greatly reduced when the absolute value of weighted mean residuals is minimized. Therefore, in addition to the benchmark regression Equation (1), five representative sub-sites of 10%, 25%, 50%, 75% and 90% were selected in this paper, and the self-help method was used to repeat 300 times for the quantile regression of agricultural green total factor productivity (see Table 5), so as to test the robustness of the benchmark regression.

According to the regression results in Table 5, it can be seen that the coefficient of the effect of rural industrial integration on agricultural GTFP is positive and passes at least the 10% significance level at different quartiles, which once again confirms the robustness of the promotion effect of rural industrial integration on agricultural GTFP. From the estimated coefficients, the absolute value of the coefficient of the impact of rural industrial integration on GTFP in agriculture increases asymptotically and then decreases as the quantile gradually increases. This indicates that the impact of rural industrial integration on agricultural GTFP has strong heterogeneity at different quantile points, and as the agricultural GTFP quantile point increases, the promotion effect of rural industrial integration shows an “inverted U-shaped” characteristic of first increasing and then decreasing. To some extent, it reflects that the promotion effect of rural industrial integration is better in regions with moderate agricultural GTFP, while the promotion effect of rural industrial integration slows down in regions with very low or very high agricultural GTFP.

#### 4.2.2. Robustness Test Based on Tobit model

The Tobit model is a semi-parameter estimation method proposed by Tobin (1958) [59]. The Tobit model does not need to assume a specific form of residuals, and can obtain a consistent estimator even in the case of individual heteroscedasticity [60]. Therefore, in order to ensure the credibility and stability of the baseline regression results, the Tobit model was used in this study to test the robustness of the baseline regression again. The general form of the Tobit model is as follows:(12)$$\left\{\begin{array}{ll} y^{*}=\beta X_{i}+\mu_{i} & \\ y_{i}=y_{i}^{*} & \mathrm{if}\;y_{i}^{*}>0\\ y_{i}=0 & \mathrm{if}\;y_{i}^{*}>0 \end{array}\right.$$

In Equation (12), $y_{i}^{*}$ is the latent variable, $y_{i}^{*}$ is the observed dependent variable, Xi is the independent variable vector, β is the correlation coefficient vector and the error term $\mu_{i}$ is independent and follows normal distribution.

Before using panel Tobit for regression analysis, appropriate model should be selected first. After passing the Hausman test, the fixed effects model is finally selected in this paper. The results showed (limited to the length and not reported in the text) that the coefficient of rural industry integration was 0.134 and significantly promoted the growth of agricultural green total factor productivity at the 5% level, which again demonstrated the robustness of the conclusion of the benchmark regression study.

### 4.3. Heterogeneity Tests

#### 4.3.1. Heterogeneity Test Based on GTFP Segmentation Index

According to the connotation of TFP growth, agricultural GTFP growth is mainly driven by agricultural technology progress and improvements in agricultural technology efficiency. The previous section confirms that rural industrial integration contributes to agricultural GTFP growth, but does this contribution come mainly through agricultural technology progress? Or is it mainly achieved through improvements in agricultural technology efficiency? The answer is still unclear. For this reason, this study further subdivides the agricultural GTFP index into the agricultural green progress index (GTC) and the agricultural technology efficiency index (GEC), and discusses the effects of rural industrial integration on the two subdivided indices separately, and the regression results are reported in Table 6.

It can be seen that the regression coefficients of rural industrial integration on the agricultural green technology progress index and the agricultural green technology efficiency index are both significantly positive, indicating that rural industrial integration helps to improve the level of agricultural green technology and promotes the improvement of agricultural green technology efficiency. In comparison, the regression coefficients of rural industrial integration on the agricultural green technology progress index are significantly stronger than those on agricultural green technology efficiency, both in terms of significance and coefficient size.

#### 4.3.2. Heterogeneity Test Based on the Level of Rural Industrial Integration

As the largest developing country in the world, provinces have large differences in agricultural resource endowment conditions and agricultural development levels, which makes the level of rural industrial integration in China uneven or uneven in development among different provinces. This may lead to a greater heterogeneity in the impact of rural industrial integration on GTFP growth in agriculture due to the high and low differences in the level of rural industrial integration.

Thus, this study takes the mean value of the median rural industrial integration index of 30 sample provinces from 2011 to 2020 as the benchmark, divides the full sample into two subsamples of high-level rural industrial integration areas and low-level areas, and examines the impact of rural industrial integration on agricultural GTFP growth under different subsamples, respectively. The regression results are shown in columns (1) and (2) of Table 7. It can be found that the coefficients of rural industrial integration in both subsamples are significantly positive, but the contribution of rural industrial integration to agricultural GTFP growth is greater in areas with higher levels of rural industrial integration.

#### 4.3.3. Heterogeneity Test Based on before and after the Rural Industrial Integration Pilot Policy

During the sample period (2011 to 2020), there were notable disparities in the weight that the state gave to the growth of rural industrial integration. Specifically, before 2015, although rural industrial integration had developed greatly throughout the country, it was more of a spontaneous behavior of economic agents, and the government did not have clear support at the policy level. In 2015, the General Office of the State Council’s “Guidance on Promoting the Integrated Development of Three Rural Industries” first proposed to promote the integrated development of three rural industries and made it an important way to promote farmers’ income and the set of modern agricultural production and management system. In 2016, the state selected 12 provinces (cities) such as Anhui and Chongqing to carry out pilot projects on the integration of three rural industries, and increased financial support for the integration of three rural industries. Additionally, in subsequent years, the Central Government’s No. 1 documents all stressed the need to support the development of the integration of three rural industries, which tries to promote the high-quality development of the agricultural economy.

Considering that national policy support has been an important factor influencing the development of Chinese agriculture, it makes the impact of the integration of the three rural industries on GTFP growth in agriculture potentially more heterogeneous depending on the changes in national policies. Therefore, this study divides the sample years into two intervals, 2011 to 2015 and 2016 to 2020, each representing a different degree of rural industrial integration policy, with 2011 to 2015 set as a low policy concern interval and 2016 to 2020 as a high policy concern interval. Sub-sample regressions were conducted to reveal the heterogeneous effects of rural industrial integration development on GTFP growth in agriculture under different levels of policy attention. The regression results are reported in column (3) and column (4) of Table 7.

It can be found that the impact coefficient of rural industrial integration is significantly positive in both sample intervals from 2011 to 2015 and from 2016 to 2020, and the impact coefficient is 0.298 in the period from 2016 to 2020, which is significantly higher than that of 0.096 in the period from 2011 to 2016. Thus, it shows that with the increase in national attention to the policy of rural industrial integration, the contribution to GTFP growth in agriculture has also expanded. Additionally, this demonstrates how efficient national policy support for rural industrial integration has great value in the expansion of agriculture GTFP.

### 4.4. Influence Mechanism Test: The Regulating Effect of Rural Human Capital or Land Transfer and Rural Industrial Integration

In this study, the interaction terms of migratory human capital, education human capital and healthy human capital or land transfer and rural industrial integration are established separately to further explore the impact of rural industrial integration on agricultural GTFP. We empirically test this theoretical hypothesis by estimating the model using the systematic GMM. As shown in Table 8, column (1) shows the interaction effect of migratory human capital and rural industrial agro-integration, column (2) shows the interaction effect of education human capital and rural industrial integration, column (3) shows the interaction effect of healthy human capital and rural industrial integration, and column (4) shows the interaction term of land transfer and rural industrial integration.

The results in columns (1) to (4) of Table 8 show that the estimated coefficients of rural industrial integration become unstable after the model incorporates three interaction terms of rural human capital or land transfer and rural industrial integration. This is because after the interaction terms of three types of rural human capital or land transfer and rural industrial integration are added, respectively, the impact of rural industrial integration on agricultural green total factor productivity changes from β2 in benchmark Equation (1) to *β2 + β5RII*_it_ in Equations (2) and (3). Here, we focus on the coefficients of the interaction term between rural industrial integration and rural human capital, i.e., the synergistic effect or joint impact mechanism of different types of rural human capital and rural industrial integration on agriculture GTFP. The interaction coefficients of migration, healthy and education rural human capital and rural industrial integration are significantly positive, indicating that the combined effect of the improvement of these three rural human capital levels and rural industrial integration will significantly promote the agriculture GTFP growth. The above results also imply that the increase in investment in rural transportation and communication, education and training, and health care under rural industrial integration can significantly promote the agriculture GTFP growth. This may be because the investment in rural transportation and communication, education and training, and health care benefits the remaining rural population, improves the interpersonal communication radius and health level of agricultural production and management entities and farmers, leads to further optimization of the supply conditions of agricultural labor factors and, thus, improves the efficiency of agricultural production and management. In addition, we also discuss the synergistic effect or common influence mechanism of land transfer and rural industrial integration on agricultural TFP, that is, the interaction coefficients of land transfer and rural industrial integration are significantly positive, indicating that the combined effect of land transfer and rural industrial integration will significantly promote the agriculture GTFP growth. This may be the development of modern agricultural operation organization caused by land transfer, which helps to better absorb modern agricultural technology, production equipment and management mode. It also helps agricultural operation entities to be embedded in all links of modern agriculture and related industrial chains, participate in specialized division of labor and market-oriented cooperation, improve agricultural production and operation efficiency, and thereby promote the growth of agricultural green total factor production.

## 5. Research Conclusions and Policy Implications

### 5.1. Research Conclusions

In this study, agricultural carbon emissions caused by six factors, including chemical fertilizer, pesticide, agricultural film, diesel oil, ploughing and irrigation, were taken as undesired agricultural output, and the agricultural GTFP growth index of China from 2011 to 2020 was measured. For the first time, the realization path of agricultural GTFP growth was discussed from the perspective of rural industry integration to provide empirical evidence and policy enlightenment for realizing carbon emission reduction, climate change governance and sustainable growth in agriculture. The results show that:(1)The integrated development of rural industries is conducive to the growth of agricultural GTFP. After the decomposition of agricultural GTFP into the agricultural green technology progress index and agricultural green technology efficiency index, it is found that the integration of rural industries can promote agricultural green technology progress and green efficiency improvement, but the promotion effect of agricultural green technology progress is more obvious. This shows that, in the context of the increasingly prominent trend of global agricultural high carbonization and the major challenges facing agricultural sustainable development, promoting the integrated development of agriculture and related industries can achieve the growth of agricultural GTFP through the progress of agricultural green technology and the improvement of factor allocation efficiency and, thus, promote the sustainable development of agriculture.(2)Quantile regression found that with the increase in agricultural GTFP, the promoting effect of rural industrial integration presented an “inverted U-shaped” feature of first growth and then decline. This indicates that when the agricultural GTFP level is low or high, the agricultural GTFP growth effect of rural industry integration is decreased, while when the agricultural GTFP level is at a medium level, the rural industry integration can promote the growth of agricultural GTFP more.(3)Heterogeneity testing shows that in areas with a higher level of rural industry integration, the growth effect of rural industry integration on agricultural GTFP is more obvious. Moreover, with the continuous improvement of the country’s emphasis on rural industry integration, the promotion effect of rural industry integration becomes more obvious. The moderating effect test showed that health, education and training, migration of rural human capital investment and rural land transfer all strengthened the promoting effect of rural industrial integration on agricultural GTFP growth to varying degrees.

### 5.2. Policy Implications

Based on the above conclusions, this paper proposes the following policy implications:(1)All countries in the world, especially the developing countries represented by China, should take the integrated development of rural industries as the path to achieve sustainable development goals in agriculture, and promote the coordinated development of agricultural economy and environment. Countries or regions should fully combine their own agricultural characteristics, take agriculture as the industrial base, modern agricultural operating entities of moderate scale as the core, interest linkage mechanism among related entities as the link, and vertical extension of agricultural industry chain, multifunctional expansion of agriculture, integration of agricultural service industry and cultivation of new agricultural forms as the means. The new industrial development mode featuring the integration of factor resources, mutual penetration of value chain and industrial cross and coordinated development within agriculture and rural secondary and agricultural industries can enhance the integrated development level of rural industries, promoting the progress of agricultural green technology and the improvement of factor allocation efficiency, so as to realize the growth of agricultural GTFP and promote the sustainable development of agriculture.(2)The agricultural GTFP growth effect of rural industry integration mainly lies in the fact that rural industry integration can effectively promote the progress of agricultural green technology and improve the allocation efficiency of agricultural factors. Countries all over the world, especially developing countries, should actively explore the knowledge and technology spillover and sharing mechanism of rural industrial integration on agriculture, promote the development level of agricultural industry and the spillover of knowledge, management and technology of relevant agricultural operating subjects, optimize the allocation of agricultural labor, land, capital, technology and management and other production factors. To improve the overall technological progress and efficiency of agriculture.(3)In the influence of rural industrial integration on agricultural GTFP, rural human capital investment and increasing land circulation are conducive to further strengthening the growth effect of rural industrial integration on agricultural GTFP. This indicates that countries in the world should enhance the coordination of policies related to rural industrial integration, rural human capital investment and land transfer. While actively promoting the integrated development of rural industries, countries in the world should also increase the investment in rural communication, medical care, education and training, so as to improve the level of rural human capital. This will help rural industry integration to play a better role in promoting agricultural GTFP. In addition, for China, which has a large population and relatively scarce land resources, land fragmentation is obvious. It is necessary to build a fair and orderly land transfer market, transfer limited and scattered land to large farming households and family farms through land transfer, or vigorously develop farmers’ cooperative organizations, so as to improve the appropriate scale, specialization and intensive land management in order to better release the rural industry integration of agricultural GTFP growth effect.

This study constructed a theoretical analysis framework of rural industry integration and agricultural GTFP, and discussed the role of rural human capital investment and rural land transfer in the influencing mechanism. By collecting indicator data related to rural industry integration and agricultural GTFP, empirical tests were conducted. This has confirmed the agricultural GTFP growth effect of rural industry integration and the role played by rural human capital investment and land transfer. For the world, especially developing countries, through developing rural industry integration, strengthening rural human capital investment and promoting agricultural land transfer, sustainable agricultural growth can be achieved and undesired output such as agricultural carbon emissions can be reduced. Coping with global climate change, that is, the accompanying environmental problems, provides a wealth of policy enlightenment.

## Figures and Tables

**Figure 1 ijerph-20-03860-f001:**
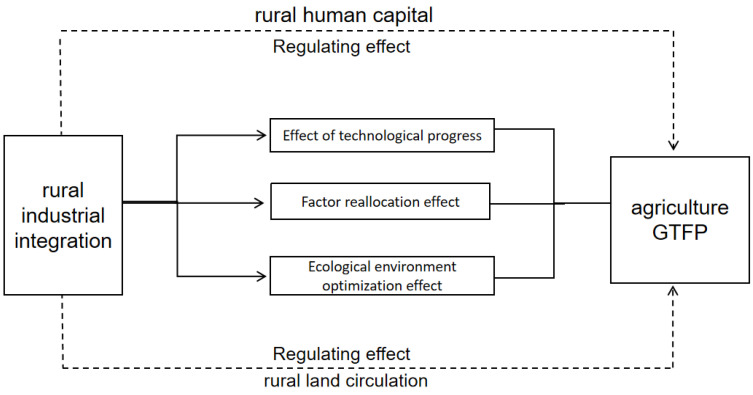
Frame diagram of the influence mechanism of rural industrial integration on agricultural GTFP.

**Table 1 ijerph-20-03860-t001:** Basis for determining agricultural carbon emission coefficients.

Carbon Emission Sources	Carbon Emission Coefficients	Reference Value Sources
Pesticides	4.9341 kg/kg	Oak Ridge National Laboratory (USA) (Li et al., 2011 [44])
Fertilizer	0.8956 kg/kg	Oak Ridge National Laboratory (USA) [46]
Diesel	0.5927 kg/kg	IPCC (Li et al., 2011; Tian et al., 2012 [44,47])
Agricultural film	5.18 kg/kg	Institute of Agricultural Resources and Ecological Environment, Nanjing Agricultural University (TIan et al., 2012 [47])
Irrigation	266.48 kg/hm^2^	(Duan et al., 2011 [48])
Tillage	312.6 kg/hm^2^	(Wu et al., 2007 [49])

**Table 2 ijerph-20-03860-t002:** Indicator system of integration of three rural industrial.

	First-Order Index	Secondary Index	Method of Measurement	Direction
Rural industrial integration development	Agricultural industry chain extension	Proportion of agricultural product processing industries	Main business income of agricultural processing industry/Total agricultural output value	+
		Scale of specialized farmer cooperatives	Number of farmers’ professional cooperatives per 10,000 people in rural areas	+
	Multifunctional expansion of agriculture	Proportion of leisure agriculture	Annual business income of leisure agriculture/Total output value of primary industry	+
		Facility agriculture level	Total area of facility agriculture/Arable land	+
		Proportion of rural non-agricultural employment	Number of people employed in secondary and tertiary industries in rural areas/Total number of people employed in rural areas	+
	Agricultural services integration	Proportion of agriculture, forestry, husbandry, fishing and service industries	Total output value of agriculture, forestry, husbandry, fishing and service industries/Total output value of primary industry	+
	Agricultural technology penetration	Degree of agricultural mechanization	Total power of agricultural machinery/Total area of arable land	+
		Agricultural labor productivity	Total output value of primary industry/Number of employees in primary industry	+

Note: + indicates a positive indicator.

**Table 3 ijerph-20-03860-t003:** Descriptive statistics of variables.

Variables	Codes	Sample Size	Mean Value	Standard Deviation	Maximum Value	Minimum Value
Green agriculture GTFP	GTFP	300	1.03	0.05	1.36	0.81
Rural industrial integration	RII	300	0.46	0.17	0.76	0.19
Level of urbanization	URB	300	0.59	0.12	0.86	0.35
Level of economic development	GDP	300	2.28	1.77	0.02	8.19
Financial support for agriculture	FINA	300	0.13	0.17	1.95	0.01
Degree of industrialization	INDU	300	0.48	0.12	0.62	0.23
Degree of openness to the outside world	OPEN	300	0.47	0.87	9.12	0.05
Digital inclusive finance	DIF	300	5.16	0.67	6.03	2.91
Migratory human capital	MH	300	11.92	2.53	18.26	7.31
Education human capital	EH	300	9.14	2.47	14.9	4.22
Healthy human capital	HH	300	8.96	2.27	17.36	4.25
Land transfer	CIR	300	0.09	0.12	0.0006	0.75

**Table 4 ijerph-20-03860-t004:** Baseline regression results.

Variables	(1)	(2)	(3)	(4)
${\mathrm{GTFP}}_{\mathrm{it}-1}$	0.061 (0.24)	0.092 * (0.73)	0.076 (0.42)	0.084 * (0.65)
RII	0.176 ** (1.82)	0.164 ** (2.31)	0.216 *** (4.22)	0.243 ** (5.71)
URB	0.152 * (4.55)	0.137 ** (3.74)	0.264 *** (5.99)	0.163 ** (4.01)
GDP	0.086 (1.24)	0.107 (1.88)	0.091 * (168)	0.088 (1.29)
FINA	0.104 *** (3.57)	0.0513 *** (2.67)	0.138 *** (4.33)	0.176 *** (6.82)
INDU	0.072 *** (4.29)	0.109 ** (5.64)	0.074 * (4.11)	0.033 * (2.38)
OPEN	0.296 * (6.01)	0.247 *** (5.22)	0.281 *** (5.64)	0.135 *** (4.28)
DIF	0.095 * (1.52)	0.108 ** (2.03)	0.077 *** (2.48)	0.058 * (2.87)
Regional effect	No	Yes	No	Yes
Time effect	No	No	Yes	Yes
AR (1)	0.01	0.02	0.01	0.01
AR (2)	0.28	0.31	0.24	0.33
Sargan	0.37	0.39	0.43	0.32
Obs	270	270	270	270

Note: ***, ** and * represent the significance level of 1%, 5% and 10%, respectively.

**Table 5 ijerph-20-03860-t005:** Robustness test based on quantile regression.

Explanatory Variable	GTFP				
	Q10 (1)	Q25 (2)	Q50 (3)	Q75 (4)	Q90 (5)
RII	0.089 ** (2.04)	0.114 ** (2.96)	0.153 ** (3.47)	0.108 *** (2.41)	0.076 * (1.85)
Regional effect	Yes	Yes	Yes	Yes	Yes
Time effect	Yes	Yes	Yes	Yes	Yes
Obs	300	300	300	300	300

Note: ***, ** and * represent the significance level of 1%, 5% and 10%, respectively.

**Table 6 ijerph-20-03860-t006:** Heterogeneity test based on each segmentation index of agricultural GTFP.

Variables	GTC (1)	GEC (2)
RII	0.168 *** (5.49)	0.083 * (2.63)
${\mathrm{GTC}}_{\mathrm{it}-1}$	0.297 *** (4.46)	
${\mathrm{GEC}}_{\mathrm{it}-1}$		0.352 *** (6.77)
Control variables	Yes	Yes
Regional effect	Yes	Yes
Time effect	Yes	Yes
AR (1)	0.08	0.06
AR (2)	0.25	0.34
Hansen	0.39	0.26
N	270	270

Note: *** and * represent the significance level of 1% and 10%, respectively.

**Table 7 ijerph-20-03860-t007:** Heterogeneity test based on the level of rural industrial integration and the degree of policy attention.

Variables	The Level of Rural Industrial Integration		Rural Industrial Integration Policy Concern Degree	
	Higher (1)	Lower (2)	2011–2015 (3)	2016–2020 (4)
${\mathrm{GTFP}}_{\mathrm{it}-1}$	0.217 *** (3.69)	0.085 ** (2.54)	0.096 *** (3.35)	0.298 *** (3.83)
RII	0.172 *** (3.77)	0.067 ** (2.62)	0.086 * (2.79)	0.163 *** (4.01)
Regional effect	No	Yes	No	Yes
Time effect	No	No	Yes	Yes
AR (1)	0.04	0.05	0.02	0.04
AR (2)	0.37	0.39	0.33	0.41
Sargan	0.27	0.32	0.35	0.31
Obs	135	135	120	135

Note: ***, ** and * represent the significance level of 1%, 5% and 10%, respectively.

**Table 8 ijerph-20-03860-t008:** The interactive effect of rural human capital or land transfer and integration of rural three industries.

Variables	(1)	(2)	(3)	(4)
${\mathrm{GTFP}}_{\mathrm{it}-1}$	0.326 *** (3.35)	0.254 *** (4.26)	0.238 *** (3.92)	0.307 *** (5.31)
RII	0.023 (0.95)	0.032 * (1.94)	0.104 (1.26)	0.175 * (2.08)
MH	0.057 * (2.46)			
RII×MH	0.918 ** (2.56)			
EH		0.211 ** (3.47)		
RII×EH		0.037 ** (3.26)		
HH			0.663 ** (4.19)	
RII×HH			0.028 * (2.58)	
CIR				0.183 *** (2.25)
RII×CIR				0.052 *** (3.79)
Regional effect	Yes	Yes	Yes	Yes
Time effect	Yes	Yes	Yes	Yes
AR (1)	0.08	0.03	0.05	0.06
AR (2)	0.24	0.19	0.34	0.16
Hansen	0.52	0.33	0.71	0.65
N	270	270	270	270

Note: ***, ** and * represent the significance level of 1%, 5% and 10%, respectively.

## Data Availability

All sample data sets are downloaded from the websites. Data are available at https://www.epsnet.com.cn/, http://rural.ccnu.edu.cn/, http://www.stats.gov.cn/, accessed on 12 August 2022.

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
