# Peer review of "Rural Industrial Integration’s Impact on Agriculture GTFP Growth: Influence Mechanism and Empirical Test Using China as an Example"

_ijerph, 2023, doi:10.3390/ijerph20053860_

Round 1
Reviewer 1 Report
Dear authors, it is interesting to use green parameters in a TFP analysis, but it is necessary to make several improvements to your paper.
1) Introduction and literature review is too long, to use so few references.
1.1) There is no clear definition of GTFP. At least construct a more finished account of Green TFP.
1.2) I recommend taking the SDGs (See for example: doi:10.1109/COMST.2018.2812301, and doi:10.1371/journal.pone.0265409.) and focus on SDG 13 and 14.
2) The theoretical analysis and research hypothesis section is excessively long in relation to the number of references used.
2.1) The way the text is presented seems to me more like a book chapter than a research article.
2.2) Compress the text and present relevant ideas or novelties.
3) The section on models, estimation methods and variables (methods) should also be more precise.
3.1) Be clear in the use of MAXDEA, in general DEA, specify better the DMUs.
3.2) Do a detailed job of justifying variables, but it does not seem to be part of the method.
3.3) Detail in the method, how it works the data, how it compares between models.
4) In the results section (Empirical Testing), do not mix methods with results. This is always confusing.
4.1) Detail methods in a previous section (Section 3 Methods).
4.2) Establish more clearly the measures of comparison between models, you must express the differences.
4.2.1) This should be clearly reflected in the tables you present in this section (Tables 4 to 8).
5) Incorporate references to your conclusions section, presenting a free text is not adequate.
6) It is important to add a discussion section.
6.1) In the discussion section, you should compare your results with the theory you present.
6.2) In the discussion section, you should show how your manuscript compares with other articles on the subject, especially recent articles.
6.3) In the discussion section, you should show how your manuscript contributes theoretically on this topic to the world knowledge in GTFP.
Author Response
The authors wish to thank the reviewer for the very helpful comments and suggestions. The manuscript has been revised according to the reviewer's suggestions and all changes have been highlighted in red in the revised manuscript. We have addressed the comments raised by the reviewer and listed the responses and modifications point by point below.
Point 1: Introduction and literature review is too long, to use so few references.
Response 1: We have put the introduction and literature review together in part 1 and added the pertinent literature. More can be seen in the red part of part 1.
Point 2: There is no clear definition of GTFP. At least construct a more finished account of Green TFP.
Response 2: In the introduction and the second paragraph of the literature review, GTFP is contrasted with TFP using references to pertinent literature, and the definition of GTFP is given. The fundamental concept of Solow's TFP is where GTFP derives, which is frequently used to gauge how well an economy is growing or developing. TFP refers to the growth rate of "desirable output" fueled by management or innovation, while disregarding material variables like labor and capital. However, it excludes "undesirable output" brought on by environmental contamination from the framework used to assess the performance of economic growth. On the other hand, Green Total Factor Productivity, which incorporates pollution emissions, an undesirable output or "bad" output, into the growth accounting framework, can more accurately reflect the genuine level of sustainable agriculture and high-quality economic development. More can be seen in the red part of part 1.
Point 3: I recommend taking the SDGs (See for example: doi:10.1109/COMST.2018.2812301, and doi:10.1371/journal.pone.0265409.) and focus on SDG 13 and 14.
Response 3: The Introduction and Literature Review in original text has described the pertinent information in the Sustainable Development Goals.
Point 4: The theoretical analysis and research hypothesis section is excessively long in relation to the number of references used.
Response 4: The quantity of references in the two chapters is balanced, the number of references in the article is reduced, and references are added to the introduction and literature review.
Point 5: The way the text is presented seems to me more like a book chapter than a research article.
Response 5: The chapter is rearranged. The article's structure is divided into six main points: the introduction and literature review, theoretical analysis and research hypothesis, model and estimation method, the choice of variables, empirical test and the discussion. The research content is thoroughly and methodically described and analyzed in each of these sections. The model, estimation technique, and variable selection are separated into two chapters in this paper. In the chapter of model and estimation method, the estimation methods involved in this paper are introduced and compared in detail. In the empirical test, only the description and analysis of regression results are kept. While in the sixth chapter, the contents of discussion are added.
Point 6: Compress the text and present relevant ideas or novelties.
Response 6: The primary contributions and overall significance of this research are discussed in the introduction and literature review. In the meantime, the research findings on the variables are reiterated at the paper's conclusion, and the policy implications and global significance are suggested. The main theme of the essay and its innovation are discussed throughout.
Point 7: The section on models, estimation methods and variables (methods) should also be more precise.
Response 7: The part of model, estimation method and variable (method) is reprogrammed, and the selection of model, estimation method and variable in this paper is divided into two chapters. In the chapter of model and estimation method, the estimation method involved in this paper is introduced, and the differences of each model and estimation method are analyzed.
Point 8: Be clear in the use of MAXDEA, in general DEA, specify better the DMUs.
Response 8: The input and output data of DMUs were analyzed comprehensively by DEA, and the quantitative indexes of the comprehensive efficiency of each DMU were derived, and the DMUs were ranked, then the effective DMUs were identified. In addition, considering the high data sensitivity requirement of MAXDEA software, the different units and great data differences among the selected variables, the input and output data were dimensionless in this paper to eliminate the differences of data of different magnitudes.
Point 9: Do a detailed job of justifying variables, but it does not seem to be part of the method.
Response 9: The justifying of variables are explained in Part 4 Variables. We provide a detailed description of why these variables were chosen and add to the relevant literature. More can be seen in Part 4 marked in red for details.
Point 10: Detail in the method, how it works the data, how it compares between models.
Response 10: The layout of the article has been reorganized to include a detailed description of both the models and estimation methods in Part 3 Models and Estimation Methods Section. And in this section several models used in this paper are compared and a more detailed description of how the estimation methods are used to process the data is provided.
In this paper, a two-stage systematic GMM estimation method is used to estimate the parameters of model (1). The standard covariance matrix of the two-step estimation method can deal with serial autocorrelation and heteroskedasticity more effectively in the usual case, and the estimation effect is more robust. However, when the distribution function of the explanatory variables is non-normally distributed, the systematic generalized moment estimation method is no longer a good estimator. At this point, the quantile regression technique is a good alternative, and quantile regression can provide a more detailed and comprehensive description of the statistical relationships between variables, and it can provide a more robust estimation. Therefore, in addition to using the benchmark regression model (1), this paper also uses quantile regression to examine the effect of rural industrial integration on GTFP in agriculture. The basic idea of quantile regression is mainly derived from Koenker and Bassett (1978), and further studied by Koenker and Hallock (2001), which focuses on the effect of each variable on the sample at different quartiles of the conditional distribution. Unlike systematic generalized moment estimation methods, quantile regression can provide multiple different regression curves, thus providing a clearer interpretation of the regression of the explanatory variables on the overall distribution and a richer amount of information to be mined. Moreover, quantile regression performs parameter estimation under the condition of minimizing the absolute value of the weighted average residuals, which greatly reduces the sensitivity to outliers and is a more robust regression. In order to better analyze the relationship between rural industrial integration and agricultural GTFP, model (4) is developed in this paper. In addition, the Tobit regression model has good advantages in dealing with truncated data, and in order to ensure the credibility and stability of the estimation results, this study chooses to use the Tobit model for further robustness testing, so model (5) is constructed in this paper.
Point 11: In the results section (Empirical Testing), do not mix methods with results. This is always confusing.
Response 11: In the empirical test section, both the empirical methods and the results are presented separately. Specifically, the empirical evidence involving the methods is placed in the methods introduction section of 3.2 and 3.3, and only the analysis of the results is retained in the empirical test.
Point 12: Detail methods in a previous section (Section 3 Methods).
Response 12: We describe all the methods covered in this paper in detail in Part 3 Model and Estimation Methods, and therefore will not repeat them in other section.
Point 13: Establish more clearly the measures of comparison between models, you must express the differences.
Response 13: The models used are described in detail in the model chapter, and the purpose and theory of each model are compared, and the points of difference between the different models are analyzed.
Point 14: This should be clearly reflected in the tables you present in this section (Tables 4 to 8).
Response 14: Tables 4 to 8 have been readjusted for a better layout.
Point 15: Incorporate references to your conclusions section, presenting a free text is not adequate.
Response 15: References are cited in the Part 6 Discussion.
Point 16: It is important to add a discussion section.
Response 16: A discussion section has been added.
Point 17: In the discussion section, you should compare your results with the theory you present.
Response 17: A discussion section was added, and the results obtained from this paper were compared with the hypotheses proposed. The results show that: after decomposing agricultural GTFP into agricultural green technological progress index and agricultural GTFP index, it is found that the promotion effect of rural industrial integration on agricultural green technological progress is more obvious, which proves hypothesis 1; the quantile regression finds that with the increase of agricultural GTFP quantile, the promotion effect of rural industrial integration shows a first increase and then decrease of "The heterogeneity test finds that the growth effect of agricultural GTFP of rural industrial integration is more obvious in areas with higher level of rural industrial integration, and the promotion effect of rural industrial integration becomes more and more obvious as the state attaches more importance to rural industrial integration; the moderating effect test finds that healthiness, education and training, migratory rural human capital investment, and rural land transfer all strengthen the promoting effect of rural industrial integration on agricultural GTFP growth to different degrees, which proves hypotheses 2 and 3.
Point 18: In the discussion section, you should show how your manuscript compares with other articles on the subject, especially recent articles.
Response 18: A comparison between related articles on the topic and this paper is added in the discussion section, and the innovative points of this paper are presented again.
Most scholars regard the pollution element of agricultural products as a kind of "bad" agricultural output and use the ML index, SBM-L index, SBM-ML, SBM-GML index and other methods to measure the GTFP of agriculture. The second is the study of factors influencing GTFP in agriculture. It is generally believed that factors such as economic development level, mechanization level, education level, natural disasters, trade level, financial support and environmental regulation affect GTFP in agriculture; the third is the study of regional differences, influencing factors and spillover effects of GTFP in agriculture from a spatial perspective. This study innovatively incorporates rural industrial integration into the framework of agricultural GTFP analysis, theoretically explains the impact and effect of rural industrial integration on agricultural GTFP growth, and further discusses the moderating role of rural human capital and land transfer in the above impact. The results of the study show that: after decomposing agricultural GTFP into agricultural green technological progress index and agricultural green technological efficiency index, it is found that the promotion effect of rural industrial integration on agricultural green technological progress is more obvious; the quantile regression finds that as the quantile of agricultural GTFP increases, the promotion effect of rural industrial integration shows a first increase and then decrease. "The heterogeneity test found that the agricultural GTFP growth effect of rural industrial integration was more obvious in areas with higher levels of rural industrial integration, and the promotion effect of rural industrial integration became more and more obvious as the national emphasis on rural industrial integration continued to increase; the moderating effect test found that healthiness, education and training, migratory rural human capital investment, and rural land transfer all strengthen the promoting effect of rural industrial integration on agricultural GTFP growth to different degrees.
Point 19: In the discussion section, you should show how your manuscript contributes theoretically on this topic to the world knowledge in GTFP.
Response 19: A theoretical contribution of the topic to world knowledge in GTFP has been added to the Discussion section. This study constructs a theoretical analysis framework of rural industrial integration and agricultural GTFP and discusses the role of rural human capital investment and rural land transfer in the influence mechanism. By collecting data on indicators related to rural industrial integration and agricultural GTFP, the empirical test confirms the effects of rural industrial integration on agricultural GTFP growth and the roles played by rural human capital investment and land transfer, which provide rich policy insights for the global, especially developing countries, to achieve sustainable agricultural growth, reduce non-consensual outputs such as agricultural carbon emissions, and cope with global climate change, i.e., many accompanying environmental and control issues through developing rural industrial integration, strengthening rural human capital investment and promoting agricultural land transfer.
Reviewer 2 Report
Referee report on “Rural Industrial Integration's Impact on Agriculture GTFP 2 Growth: Influence Mechanism and Empirical Test—Using 3 China as an Example”
The article analyzes the dynamics of the GTFP for Chinese provinces, decomposing the contribution of several relevant factors and variables to the evolution of this component through various econometric methods and robustness analyses.
In general, I believe that the work is very well motivated, the presentation is quite clear, and within the limits of the quality of the available data, the results presented seem robust and the econometric methods are used properly, and the various robustness analyzes presented in the article strongly support the conclusions of the work. So I believe that the work has quality and merit for publication in IJERPH, and has the potential for a work of great impact. In general, I have no additional recommendations regarding the motivation, presentation or methods used in the article.
My only recommendation would be to add in section 3.2 a discussion about possible limitations on the quality of the data used and the possible impacts of measurement errors on the analysis (which are potentially already addressed by the System GMM method used). Although the authors use the better existing data, and the work of constructing variables is adequately explained, it is important to bring a discussion of limitations in existing data into the analysis.
I believe that the work has a very relevant contribution, the analyses are carried out properly and the presentation is very clear. So my recommendation is for a minor revision, based on my suggestions.
Specific comments:
Abstract – defines GTFP
Abstract - systematic GMM to system GMM
Page 2 – defines the concept of GTFP
page 6, line
Section 3.2 - Add a brief discussion on the quality of measurement of the variables used in the estimation.
Author Response
The authors wish to thank the reviewer for the very helpful comments and suggestions. The manuscript has been revised according to the reviewer's suggestions and all changes have been highlighted in red in the revised manuscript. We have addressed the comments raised by the reviewer and listed the responses and modifications point by point below.
Point 1: My only recommendation would be to add in section 3.2 a discussion about possible limitations on the quality of the data used and the possible impacts of measurement errors on the analysis (which are potentially already addressed by the System GMM method used). Although the authors use the better existing data, and the work of constructing variables is adequately explained, it is important to bring a discussion of limitations in existing data into the analysis.
Response 1: Combined with the relevant literature, the concept of GTFP is defined in the second paragraph of the Introduction and Literature Review, and the variable measurement quality is synthesized by applying the SBM-ML index method to the data indicators of rural industrial integration and agricultural total factor productivity in the selection of variables in Part 4, mainly referring to the relevant literature such as Sun (2022), Hu (2022), Li (2011), and Tian (2012) to ensure the reliability of the data at a certain level.
Round 2
Reviewer 1 Report
The manuscript continues to be excessively long and provides little new in each section.
It incorporates the Green concept into Total Factor Productivity without a clear and precise justification regarding the paradigm shift proposed by the SDGs (initially SDG 13 and now also SDG 14).
Keeping a theoretical analysis and then an empirical one, it seems to present 2 articles in one. Generating a confusing structure of the manuscript.
Using DEA without clearly defining the DMU (Decision Making Unit) that are evaluated does not make sense either.
The authors must make these corrections before they can publish this paper.
Author Response
The authors wish to thank the reviewer for the very helpful comments and suggestions. The manuscript has been revised according to the reviewer's suggestions and all changes have been highlighted in red in the revised file. We have addressed the comments raised by the reviewer and listed the responses and modifications point by point below.
Point 1: The manuscript continues to be excessively long and provides little new in each section.
Response 1: We have compressed the introduction and literature review sections. Firstly, in the Introduction and Literature Review, the common knowledge and general descriptions were deleted, and the relevant introduction of the United Nations 2030 Agenda for Sustainable Development regarding climate change was re-added in conjunction with the topic of this study, which further highlights the significance of the selected topic of this study. Second, because it overlaps with the second section, the material on rural human capital investment and rural land transfer was removed. This section overlaps with the theoretical analysis section of the article, which has provided a detailed explanation of the regulation mechanism governing rural human capital investment and rural land transfer in relation to the effects of rural industrial integration on agricultural GTFP.
It is also important to note that the manuscript is still relatively long compared with most similar articles even after being condensed. The reasons are (1) The theoretical analysis section of this study not only explains how rural industrial integration affects agricultural GTFP, It also examines the moderating effects of rural human capital investment and rural land transfer on the aforementioned influence. The analysis of the moderating effects is a significant theoretical innovation or contribution of this study. (2) We had to create a multidimensional indicator system to measure the connotation of rural industrial integration in the model, variables, and data description part (Part III) because the primary explanatory variable of this study, rural industrial integration, has no published official data. Agricultural GTFP, the study's explanatory variable, still needs to be incorporated to the measurement process, indicators, and data sources. We have introduced the pertinent measurement techniques, indicators, and data of rural industrial integration and agricultural GTFP in detail in order to emphasize the readability and logical soundness of this work. This significantly expands the space. (3) This study discusses the heterogeneity of rural industrial integration affecting agricultural GTFP from various aspects such as agricultural GTFP decomposition indices (agricultural technical progress index and agricultural technical efficiency index), different regions, and different time periods in the empirical test section in addition to the benchmark regression and robustness test. Additionally, it experimentally investigates the moderating effects of two factors, namely rural land transfer and investment in human capital, on agricultural GTFP during the process of rural industrial integration. For these reasons, the paper is relatively long.
Point 2: It incorporates the Green concept into Total Factor Productivity without a clear and precise justification regarding the paradigm shift proposed by the SDGs (initially SDG 13 and now also SDG 14).
Response 2: The context of SDG 13 (primarily climate change) and agricultural GTFP in the United Nations 2030 Agenda for Sustainable Development were introduced or explained in the first and second paragraphs of the introduction, respectively. In contrast, non-consensual output in the measurement of agricultural GTFP is defined as agricultural carbon emissions brought on by six causes, including fertilizer, pesticide, agricultural film, diesel, tillage, and irrigation (Li et al., 2011; Tian et al., 2012). As a result, the UN 2030 Agenda for Sustainable Development's target 13, which suggests an action plan to combat climate change, and the growth of the agricultural GTFP are compatible.
Point 3: Keeping a theoretical analysis and then an empirical one, it seems to present 2 articles in one. Generating a confusing structure of the manuscript.
Response 3: As an empirical research article, the manuscript should first undertake a theoretical analysis, construct the research hypotheses to be demonstrated, and then gather sample data for empirical testing. It draws from similar empirical research literature. In addition to discussing the moderating effects of rural human capital investment and rural land transfer in the aforementioned impact, this study focuses on the effect of rural industrial integration on agricultural GTFP. Because of this, we focused on the theoretical analysis of the aforementioned problems in the second part. And during the revision process, we created a theoretical analysis framework diagram (see Figure 1 in the paper) for the reader's comprehension. The model, primary indicators, and data sources of this study are also introduced in the third section. While in the fourth part, the empirical tests are conducted for the theoretical analysis and research hypotheses in the second part. We discuss the study's findings and their implications for policy in the fifth section.
Point 4: Using DEA without clearly defining the DMU (Decision Making Unit) that are evaluated does not make sense either.
Response 4: In the revised manuscript of 1st round, we have added measures of DMU and agricultural GTFP drawing on existing literature.